# Comparison of Secular Trends in Peptic Ulcer Diseases Mortality in China, Brazil and India during 1990–2019: An Age-Period-Cohort Analysis

**DOI:** 10.3390/healthcare11081085

**Published:** 2023-04-11

**Authors:** Jinyi Sun, Lihong Huang, Ruiqing Li, Tong Wang, Shuwen Wang, Chuanhua Yu, Jie Gong

**Affiliations:** 1School of Public Health, Wuhan University, Wuhan 430071, China; 2Global Health Institute, Wuhan University, Wuhan 430071, China; 3Wuhan Municipal Center for Disease Control and Prevention, Wuhan 430024, China

**Keywords:** peptic ulcer disease (PUD), mortality, age-period-cohort (APC) model

## Abstract

Background: Peptic ulcer disease (PUD) is a common disease worldwide, especially in developing countries. China, Brazil, and India are among the world’s fastest-growing emerging economies. This study aimed to assess long-term trends in PUD mortality and explore the effects of age, period, and cohort in China, Brazil, and India. Methods: We collected data from the 2019 Global Burden of Disease Study and used an age–period–cohort (APC) model to estimate the effects of age, period, and cohort. We also obtained net drift, local drift, longitudinal age curve, and period/cohort rate ratios using the APC model. Results: Between 1990 and 2019, the age-standardized mortality rates (ASMRs) of PUD and PUD attributable to smoking showed a downward trend in all countries and both sexes. The local drift values for both sexes of all ages were below zero, and there were obvious sex differences in net drifts between China and India. India had a more pronounced upward trend in the age effects than other countries. The period and cohort effects had a similar declining trend in all countries and both sexes. Conclusions: China, Brazil, and India had an inspiring decrease in the ASMRs of PUD and PUD attributable to smoking and to period and cohort effects during 1990–2019. The decreasing rates of *Helicobacter pylori* infection and the implementation of tobacco-restricting policies may have contributed to this decrease.

## 1. Introduction

Peptic ulcer disease (PUD) is defined as damage to the digestive tract with mucosal ruptures greater than 3–5 mm making the submucosa visible [1,2]. PUD is still a common condition worldwide [3]. In the general population of Western countries, the lifelong prevalence of PUD is approximately 5–10%, with an annual incidence of 0.1–0.3% [1,2,4]. Complications of PUD, such as gastrointestinal bleeding and perforations in the gastrointestinal tract, also have a high mortality rate. Globally, the number of hospitalizations for peptic ulcer bleeding has steadily decreased, but the case fatality rate has stabilized at 5–10% [5]. A perforation usually manifests as a sudden attack of severe epigastric pain. Depending on variations in complications and age, the death rate can be as high as 20% [6,7]. The burden of disease caused by PUD is very high. The annual healthcare organization expenditure attributable to PUD per capita was USD 1183, and the total healthcare organization expenditure attributable to PUD was USD 59.4 million per year [8].

Both Helicobacter pylori (HP) infection and the use of non-steroidal anti-inflammatory drugs (NSAIDs) are major aetiological factors for PUD and peptic ulcer complications [9,10]. The prevalence of HP infection is around 50% worldwide, with the highest rates present in developing nations [11]. It is most common in developing countries and among those in lower socio-economic and educational groups [12]. Smoking causes a wide range of diseases, including many kinds of cancer, chronic obstructive pulmonary disease, and PUD [13]. In developed countries, major campaigns have been launched to reduce smoking rates. In contrast, the public in most developing countries is not aware of the dangers [14].

China, Brazil, and India share similar characteristics, including vast territories, dense populations, and abundant resources. As these three fast-growing developing countries (China, Brazil, and India) are similar, they launched BRICS with Russia and South Africa, which aims to enhance cooperation between these countries. PUD is a common concern in China, Brazil, and India. The prevalence of endoscopically confirmed PUD is higher in the population of Shanghai (17.2%) than in Europe (4–6%) [15]. The prevalence of duodenal ulcers and HP infections in the Brazilian population is high [16]. A retrospective analysis in Porto Alegre, Brazil, showed that the prevalence of PUD in Brazil was decreasing annually, from 8.6% in 1996 to 3.3% at the end of 2005. A study of the general population of Kashmir, India, showed a point prevalence of 4.72% and a lifelong prevalence of 11.22% for peptic ulcers [17]. Virendra Singh surveyed a community in northern India and found a point prevalence of 3.4% and a lifelong prevalence of 8.8% for active peptic ulcers [18].

BRICS accounts for about 25% of the world’s gross national income, more than 40% of the global population and nearly 40% of the global illness burden [19,20]. Additionally, in 2017, BRICS reaffirmed its efforts to achieve the 2030 Sustainable Development Goals [21]. Progress in health policy and PUD prevention and control in BRICS has guiding implications for low- and middle-income countries [22]. It is worth comparing China, Brazil, and India, as they represent three of the developing countries included in BRICS. This comparison can help us understand the general situation of PUD in developing countries and formulate corresponding policies to reduce the prevalence of PUD.

We aim to analyze a 30-year trend in PUD and PUD attributable to smoking mortality by exploring the effects of age, period, and cohort in China, Brazil, and India. This analysis can assist governments in formulating targeted measures to meet the needs of their respective populations. Analyzing data from China, Brazil, and India can help us achieve the 2030 goal of reducing non-communicable diseases (NCDs) early. The age, period, and cohort (APC) model has been widely used to analyze the mortality trends of NCDs because it can demonstrate the effects of age, period, and cohort. We used the 2019 Global Burden of Disease Study (GBD) data to conduct an APC analysis of PUD in China, Brazil, and India. This is the first comprehensive report on an APC analysis of PUD in China, Brazil, and India in 30 years.

## 2. Methods

### 2.1. Data Sources

Based on systematic and standardized estimations of 369 diseases and injuries from 1990 to 2019, GBD 2019 provides data from 204 countries and territories [23]. GBD 2019 provided population, deaths due to PUD, and age-standardized mortality rates (ASMR) for China, Brazil, and India from 1990 to 2019, including different sexes and age groups. Their ASMR were standardized to the GBD 2019 global age-standard population. ASMR calculations are made using the following formula:$$ASMR=\frac{\sum \begin{array}{c} Age\;composition\;of\;standard\;group\;population\\ \times Age\;specific\;mortality \end{array}}{Age\;composition\;of\;standard\;population}$$

The population attributable fraction (PAF) is defined as the proportion of related diseases or deaths in a population that would be reduced if exposure to a certain risk factor was lowered to its theoretical minimum exposure level [24]. In the GBD 2019 study, the theoretical minimum exposure level for smoking was set to 0 [25]. We estimated the PAF for PUD mortality attributable to smoking by combining the distribution of smoking exposure with exposure risk estimates for each level of exposure:$$PAF=\frac{p\left(n\right)+p\left(f\right)\int \exp\left(x\right)*rr\left(x\right)+p\left(c\right)\int \exp\left(y\right)*rr\left(y\right)-1}{p\left(n\right)+p\left(f\right)\int \exp\left(x\right)*rr\left(x\right)+p\left(c\right)\int \exp\left(y\right)*rr\left(y\right)}$$
where the prevalence of never smokers, former smokers, and current smokers is represented by 𝑝(𝑛), 𝑝(𝑓), and 𝑝(𝑐) respectively. The distribution of years since quitting among former smokers is represented by exp(𝑥), while the relative risk for years since quitting is represented by 𝑟𝑟(𝑥). The distribution of cigarettes per smoker per day or pack-years is represented by exp(𝑦), and the relative risk for cigarettes per smoker per day or pack-years is represented by 𝑟𝑟(𝑦). Cigarettes per smoker per day was used for PUD and cardiovascular diseases. The number of PUD deaths attributed to smoking can be calculated by multiplying the number of deaths due to PUD by the PAF of PUD attributed to smoking [26]. The ASMR of PUD attributed to smoking was calculated using the GBD 2019 global standard population.

The cause of death ensemble model (CODEm) is a cause of death combinatorial modeling tool that can estimate cause-specific mortality in different places, including age and sex [27]. DisMod-Mr2.1 is a Bayesian meta-regression tool developed specifically for GBD and is widely used to generate prevalence and estimate incidence [28]. By using CODEm and DisMod-Mr2.1, vital registration, verbal autopsy, registry, survey, police, and surveillance data comprise the original mortality database [29,30].

There are some ways to obtain original death data in China, Brazil, and India: for China, this mostly includes the Cause of Death Reporting System, Disease Surveillance Points [31]; for Brazil, this mostly includes the Mortality Information System [32]; for India, this mostly includes the Registrar General and Sample Registration System [33].

### 2.2. Statistical Analysis

The APC Web Tool is used for parameter estimation, along with associated statistical hypothesis tests. We adopted this tool, which can be accessed here: http://analy-sistools.nci.nih.gov/apc/( accessed on 26 October 2022).

Because those who died from PUD aged <15 years were few, they were not considered in this study. From 1990 to 2019, a series of consecutive 5-year periods were considered appropriate for classifying the ASMR of PUD. So, we obtained 5-year age groups from 15–19 years to 80–84 years, and the corresponding consecutive 19 birth cohorts (from 1910–1914, 1915–1919, etc. to 2000–2004). In addition, we referred to a central age group (45–49 years old), the 2000–2004 period, and the cohort of 1955–1959.

Since birth cohorts were calculated according to the death time period and death age of the individual (i.e., birth cohorts = period − age), we used the APC model to assess a relationship between age, period, and birth cohort and PUD mortality [34]. Because of the linear relationship among age, period, and cohort, identification problems may still exist in the APC model. By using the intrinsic estimation method based on estimable function and matrix singular value decomposition, we can solve collinearity problems [35].

Period effects are changes with the passage of time that affect each age group at the same time and may be due to changes in the social, cultural, economic, or natural environment. Cohort effects are associated with changes between groups of persons born in the same year. Net drift shows the holistic annual percentage change in the expected age-adjusted ratio over time, expressed as an integral log-linear trend between period and cohort. Local drift represents the annual percentage change in the expected age-adjusted ratio over time for each group, indicating a partial log-linear trend between period and cohort. The longitudinal age curve reflects the fitted age-specific rates in the reference cohort adjusted for period effects. The cross-sectional age curves depict the predicted age-specific rates in the reference period after accounting for the cohort influence. Period and cohort relative risk are the ratio of the age-specific ratio for each corresponding period and cohort relative to the reference. 

The Wald Chi-square test was performed to verify the importance of estimable parameters and functions (Appendix A). We used a general linear model to assess the interaction terms between sex and calendar year/birth cohort and to evaluate the slope of period/cohort RRS for statistical significance. The standard APC model was expressed as follows:$$\rho =\log\left(\theta_{apc}/N_{apc}\right)=\alpha_{a}+\pi_{p}+\gamma_{c}$$
where the log transformation of the expected mortality rate of PUD can be illustrated by ρ. Age, period, and cohort effects can be expressed as α, π, and γ, respectively.

The longitudinal form and cross-sectional form of the APC model could be expressed as:$$\rho_{ac}=\mu +\left(\alpha_{L}+\pi_{L}\right)\left(a-\bar{a}\right)+\left(\pi_{L}+\gamma_{L}\right)\left(c-\bar{c}\right)+\tilde{\alpha_{a}}+\tilde{\pi_{p}}+\tilde{\gamma_{c}}$$
$$\rho_{ap}=\mu +\left(\alpha_{L}-\gamma_{L}\right)\left(a-\bar{a}\right)+\left(\pi_{L}+\gamma_{L}\right)\left(p-\bar{p}\right)+\tilde{\alpha_{a}}+\tilde{\pi_{p}}+\tilde{\gamma_{c}}$$
where the longitudinal age pattern is illustrated by ($\alpha_{L}+\pi_{L}$), the cross-sectional age pattern can be shown by ($\alpha_{L}-\gamma_{L}$), the net drift is represented by ($\pi_{L}+\gamma_{L}$), and $\tilde{\alpha_{a}}$, $\tilde{\pi_{p}}$, and $\tilde{\gamma_{c}}$ are the deviations for age, period, and cohort, respectively.

### 2.3. Ethics Consideration

Our study was based on a publicly available GBD database. No patients, public, or animals were involved in the design, conduct, reporting, or dissemination plans of our study. The data mentioned above were available here: http://ghdx.healthdata.org/GBD-resultstool (accessed on 26 October 2022). Therefore, ethical approval is not required for our study.

## 3. Results

### 3.1. Trends in Deaths and Age-Standardized Mortality Rates of PUD

The trends in PUD mortality across China, Brazil, and India are presented in Table 1. Between 1990 and 2019, China, Brazil, and India all demonstrated meaningful decreases in the ASMRs of PUD in both sexes (*p* < 0.01). The absolute number of deaths from PUD decreased in all countries for both sexes, except for Brazilian and Indian women (1724 to 2103 and 32,743 to 34,191, respectively). Compared to 1990, the relative proportions of all-cause deaths due to PUD in China, Brazil, and India decreased to varying degrees for all sexes. Despite the population increasing globally, China’s percentage of all-cause deaths due to PUD compared to the global population decreased for both sexes between 1990 and 2019 (22.65% to 18.68% for men; 21.59% to 18.09% for women).

From 1990 to 2019, India had the largest decrease in PUD ASMRs, from 20.17/100,000 in 1990 to 6.70/100,000 in 2019 (−66.78%), with the fastest annual percentage change (−4.34%). For Indian men, PUD ASMRs decreased by 69.84%, and the annual percentage change was up to −4.67%, while in 2019, the ASMRs of PUD in India were still much higher than in Brazil and China for both sexes. The relative proportions of all-cause deaths due to PUD in China represented the most significant decreases for both sexes (−46.1% for men, −38.9% for women).

### 3.2. Age-Standardized Mortality Rates of PUD Attributable to Smoking

The ASMRs of PUD attributable to smoking by sex across China, Brazil, and India during 1990–2019 is illustrated in Figure 1. On the whole, China, Brazil, and India show a similar downward trend from 1990 to 2019, with significant sex differences (the ASMRs of all countries were significantly higher in men than women). Between 1990 and 2019, Indian men showed the most dramatic reduction (dropping by 74.3%, from 6.33/100,000 to 1.63/100,000); Chinese and Brazilian men also displayed impressive improvements in the ASMRs of PUD attributable to smoking (dropping by 67.0% from 4.49/100,000 to 1.48/100,000 and dropping by 72.9% from 3.43/100,000 to 0.93/100,000, respectively). In contrast, women in China, Brazil, and India showed a relatively stable declining trend (from 0.32/100,000 to 0.11/100,000, 1.20/100,000 to 0.35/100,000, and 0.83/100,000 to 0.29/100,000, respectively). In general, the ASMRs of PUD attributed to smoking exhibited a faster declining trend in all countries and both sexes in the period of 1990–2012, but this reduction tended to stabilize in the following year.

### 3.3. Net Drifts and Local Drifts for PUD Mortality

The net drifts and local drifts for PUD mortality in China, Brazil, and India by sex from 1990 to 2019 are shown in Figure 2. In the case of Brazil, men (−3.93%/y [95%CI, −4.04 to −3.82]) and women (−3.82%/y [95%CI, −4.00 to −3.64]) had a similar declining trend, whereas, for China and India, the net drifts presented significant sex variations. In the case of China, men’s mortality decreased to a lesser degree than women’s mortality (−4.97%/y [95%CI, −5.12 to −4.82] versus −6.12%/y [95%CI, −6.45 to −5.79]). For India, men’s mortality decreased to a greater degree than women’s mortality (−5.12%/y [95%CI, −5.25 to −4.98] versus −4.19%/y [95%CI, −4.38 to −4.00]) (*p* < 0.01 for all).

During 1990−2019, all local drift values were less than zero for all age groups (15–85) in all countries and both sexes, suggesting significant decreases in PUD mortality (Appendix A). In the case of Brazil, the local drift values decreased rapidly in both sexes aged 15 to 34 years (−2.63% to −4.68% for men, −3.43% to −4.67% for women), increased rapidly in the 35–54 age group (−4.68% to 3.78% for men, −4.67% to −3.33% for women), and then increased slightly in both sexes aged 55–84 years. China and India presented similar trends in men of all age groups (increasing rapidly in the age group of 15–39, decreasing slightly at the ages of 40–54 and then increasing rapidly after the age of 55) (Figure 2A). For Indian women, the local drift values increased very rapidly in the age group of 15 to 39 years of age (−6.56% to −4.06 %). Then, it tended to be stable after the age of 50. Conversely, for Chinese women, the local drift values decreased rapidly until the age of 29 (−6.51% to −7.69%) and then increased monotonically in the age group of 30 to 84 years of age (−7.69% to −3.25%) (Figure 2B). This suggests that the improvements in the mortality of PUD are decreasing year by year.

### 3.4. The Age, Period and Cohort Effects on PUD Mortality

The longitudinal age curves of sex-specific PUD mortality are depicted in Figure 3. The PUD mortality rates per 100,000 increased prominently with age in all countries and both sexes, thus indicating that age is a risk factor for PUD. In the same birth cohort, the rates of China and Brazil increased monotonically in both sexes for all age groups (from 0.81 for men and 0.80 for women at 15-19 years of age to 18.45 for men and 8.85 for women at 80-84 years of age vs. from 0.60 to 15.91 in men and 0.28 to 11.20 in women in the respective age groups). Remarkably, India’s curve was steeper and higher for both sexes, especially for Indian women in the age group of 40 to 84 years of age (5.38/100,000 to 38.90/100,000).

As shown in Figure 3, the period effects on PUD mortality exhibited a similar monotonic declining trend in all countries and both sexes during 1990–2019, with China having the greatest reduction of mortality in both sexes (decreasing from 1.65 to 0.45 in men and from 1.84 to 0.37 in women). Meanwhile, the calculated cohort’s RRs for both sexes in all countries showed continuous declining trends, specifically for Indian men (decreasing from 8.37 in 1910 to 0.07 in 2000) and Chinese women (decreasing from 7.84 in 1910 to 0.05 in 2000). It is notable that Brazil presented the lowest degree of reduction in both sexes (4.73 to 0.20 in men, 4.67 to 0.16 in women). Improvements in PUD risks were the most prominent in all countries and both sexes for those born prior to 1955.

### 3.5. Age- and Cohort-Specific Mortality Rates for PUD

We divided the mortality and population data into five-year periods, ranging from 1990 to 1994 (median, 1992) and from 2015 to 2019 (median, 2017). Additionally, we obtained 19 successive cohorts, ranging from 1910 to 1914 (median, 1912) and from 2000 to 2004 (median, 2002). The age-specific mortality rates of PUD by period and sex in China, Brazil, and India during 1990–2019 are illustrated in Figure 4. Figure 4 plots a falling trend in PUD mortality between the years 1990 to 1994 and 2015 to 2019 as well as a PUD mortality rate that increases with age in all countries and both sexes (*p* < 0.01 for all). India has the most significant reduction in mortality. After controlling for cohort effects, cross-sectional age curves show the anticipated age-specific rates during the reference period, i.e., 2000 to 2004. (*p* < 0.01 for all).

The cohort–specific mortality rate of PUD by age group among China, Brazil, and India during 1990–2019 is presented in Figure 5. Brazil (Figure 5(B1,B2)) and India (Figure 5(C1,C2)) show declining trends for the PUD mortality rate, whereas China (Figure 5(A1,A2)) first showed a drop and then presented a rise and a subsequent decrease in all age groups, suggesting a relatively higher risk of PUD mortality in those cohorts born earlier (*p* < 0.01 for all). As in the example of Brazilian women, the PUD mortality of those who were 80 to 84 years of age (median, 82 years of age) gradually increased within their birth cohort (*p* < 0.01 for all).

## 4. Discussion

### 4.1. Main Risk Factors for PUD

HP infection, NSAID use, smoking, and age play major roles in the pathogenesis of peptic ulcerations [36]. The primary risk factors of PUD include HP and the widespread use of NSAIDs [9,37,38]. NSAID users’ risk of developing PUD is five times higher than that of the general population, while aspirin users’ risk is three times higher [9,39,40]. In addition to the use of NSAIDs and contracting an HP infection, the main complicating factors of the disease are individual comorbidities and aging [41,42,43]. Cigarette smoking is an etiologic factor and is coupled with the initiation, prolongation, and recurrence of gastric ulcers [44,45]. A definite positive association exists between smoking and peptic ulceration, and a dose-response effect also exists between the number of cigarettes smoked and gastroduodenal lesions [46]. An attributable risk analysis showed that 43–63% of duodenal ulcer deaths in men were caused by smoking, compared to 25–50% in women [47]. Recently, smoking has been shown to cause gastric mucosal injuries in the stomach [48]. Moreover, smoking increases the failure rate of HP eradication therapy, and the risk of HP eradication failure in smokers increases if they are current smokers and if their smoking dose is high [49,50].

### 4.2. Comparative Analysis of the Reasons for the Decrease in PUD Mortality in China, Brazil and India

During 1990–2019, there were significant improvements in PUD mortality in China, Brazil, and India. India had the greatest degree of reduction in the ASMRs of PUD (−67%), especially in men, who showed the most obvious decrease (−70%). Additionally, the relative rates of all-cause deaths due to PUD fell by more than 25% for all three countries and both sexes, except for Brazilian and Indian women (−4.65% and −7.14%, respectively). In general, China, Brazil, and India show a continuous rising age effect as well as similar decreasing period and cohort effects. Both improvements in the period and cohort effects are likely due to the contributions of improved living conditions, socioeconomic status, and better hygiene to the decrease in the prevalence of HP infection [51,52], with the successful implementation of tobacco-restricting policies in developing countries also playing a role [53,54].

India presented the most obvious decrease in PUD ASMRs among the three countries. These improvements were due to both period and cohort influences, especially for those born after 1955. The epidemiology of PUD in India changed from 1992 to 2012, including a decrease in PUD incidence frequency [51]. The prevalence of PUD in Indians steadily decreased during 1996–2016, and this reduction was synchronous with the reduction in PUD associated with HP [55]. India signed the Framework Convention on Tobacco Control (FCTC) in 2003, and legislation was introduced in 2008 to ban smoking in all public places [56,57]. In India, the overall prevalence of tobacco smoking decreased from 19.8% to 8.6% during 1987–2016; the prevalence in men varied between 36% and 16%, whereas the prevalence in women never exceeded 3% [58]. This helps to largely explain why India demonstrated the most outstanding decreases in the ASMRs of PUD attributable to smoking. Additionally, the prevalence of tobacco smoking in men in India reduced to a greater degree than in women. This could explain why PUD mortality in Indian men decreased to a greater degree than in women.

China has also seen a significant downward trend in PUD ASMRs, and these gains were the result of both period and cohort effects. The prevalence of HP among mainland Chinese adults was 49.6% (95% CI, 46.9–52.4%), and it decreased by −0.9% (95%CI, −1.1–0.6%) per year, with an annual percentage change of −1.0% (95%CI, −1.4–0.6%) for men and −1.3% (95%CI, −1.8–0.9%) for women from 1983 to 2018 [59]. Notably, women had a faster rate of decrease than men, which could be a reason for PUD mortality decreasing to a greater degree in Chinese women than in men. Shuai Ren et al. also performed a systematic review in mainland China from 1990 to 2019, which showed the prevalence of HP significantly decreasing from 58.3% (95%CI, 50.7–65.5%) in the period of 1983–1994 to 40.0% (95%CI, 38.2–41.8%) in the period of 2015–2019 [60]. China ratified FCTC in 2005, and the FCTC came into force in China on January 9, 2006 [61]. Additionally, smoking rates in China have decreased for both men and women since 1990 [62]. Chinese and Indian authorities have made efforts to limit access to e-cigarettes in recent years [63]. All of the above may have contributed to the improvements in the ASMRs of PUD and the ASMRs of PUD attributable to smoking in China. Efforts in medical insurance and technology, along with instilling regular healthy eating habits in the lives and minds of people in China have likely contributed to these improvements from 1990 to 2019 [64].

Brazil had the most minor improvement in PUD mortality and a slightly increased number of PUD deaths. In addition, Brazil also had the smallest decrease in period and cohort effects; one reason may be that the prevalence of HP in Brazil was not effectively controlled. The time trends in HP prevalence in Brazil were 68.2% from 1970 to 1999 and 71.3% from 2000 to 2016 [65]. Brazil’s achievements are very encouraging and the country’s practices are worthy of study. In 1994, the Family Health Program (FHP) was launched and extended throughout Brazil; the FHP and the following measures have benefited from the prevention, diagnosis, and treatment of NCDs [66]. The subversive implementation of primary healthcare in Brazil may have played a role, with multidisciplinary teams supporting FHP and expanding preventive and integrated care to manage NCDs [67]. Brazil established the FCTC in 2006, the Smoke-Free Law in 2014, and other tobacco-restricting laws. From 1998 to 2013, tobacco use decreased by about 60 percent [68]. This could explain why Brazil had remarkable reductions in the ASMRs of PUD attributable to smoking during 1990–2019 (−72.9%). Despite these advances, the rapidly aging population and a decrease in healthy diets leave little room for pride [67]. At the same time, the number of PUD deaths kept rising from 1990 to 2019, partly due to the aging of the population and population growth.

### 4.3. Limitations of Our Study

There are certain limitations to our study. Firstly, while we have evaluated period and cohort influences, the GBD data were not from a cohort study. Additionally, the APC analysis originated from estimated GBD cross-sectional data during 1990–2019. Large cohort studies need to be conducted in different countries to determine the relative risks of a particular location and time. Secondly, statistical objects do not include data on people under the age of 15. This is because there are few deaths from PUD under the age of 15 in China, Brazil, and India. Data on NCDs are frequently scarce and incomplete in low- and middle-income nations. Additionally, when gastrointestinal diseases are confirmed as the cause of death, they are usually not believed to be the root cause [69]. Thirdly, an ecological fallacy may have occurred since the APC model is built on a population level, which means that it might not be applicable on an individual level.

## 5. Conclusions

Although the mortality rates of PUD have decreased significantly in China, Brazil, and India, the burden of PUD is still heavy. During 1990–2019, China, Brazil, and India presented a declining trend in the ASMRs of PUD and the ASMRs of PUD attributable to smoking in both sexes, while men were at a higher risk than women. Between 1990 and 2019, the period and cohort effects decreased in all countries and both sexes. Combining the age effect, in general, men and the elderly were high-risk populations for PUD mortality. In China, Brazil, and India, there were significant reductions in PUD mortality across all age groups in both sexes over time. Additionally, in each cohort, the older age groups had higher PUD mortality among the three countries and both sexes.

India stands out for its improvements in the ASMRs of PUD and the ASMRs of PUD attributable to smoking, which may be associated with effectively reducing the prevalence of smoking tobacco. China and Brazil also had significant reductions in the ASMRs of PUD and the ASMR of PUD attributable to smoking; the implementation of tobacco-restricting policies could be one of the reasons. For China, another reason for the improvements in the ASMRs of PUD may be due to the excellent control of the prevalence of HP. The relatively minor improvements in the cohort effects in Brazil in both sexes for all age groups may be due to Brazil’s extremely high HP prevalence. These examples illustrate that it is necessary to control the rates of HP infection and the prevalence of tobacco. Additionally, Brazilian and Chinese policymakers should pay more attention to the implementation of tobacco-restricting policies and the reduction of the prevalence of HP.

## Figures and Tables

**Figure 1 healthcare-11-01085-f001:**
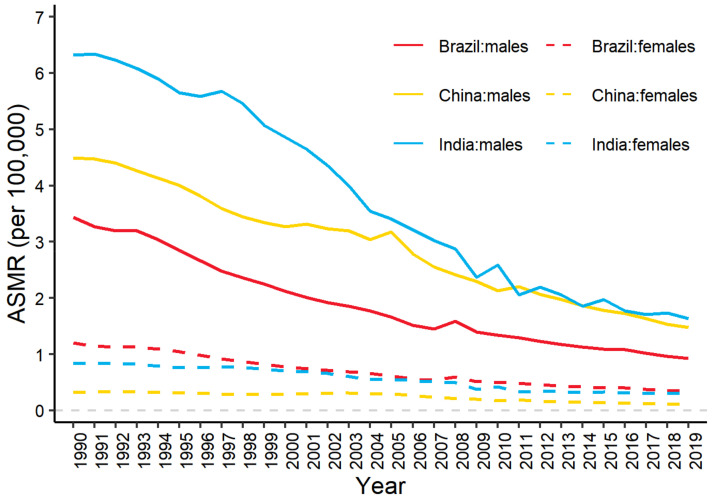
Trends in ASMR per 100,000 population of PUD attributable to smoking by sex in China, Brazil, and India during 1990–2019: standardized to the GBD 2019 global age-standard population.

**Figure 2 healthcare-11-01085-f002:**
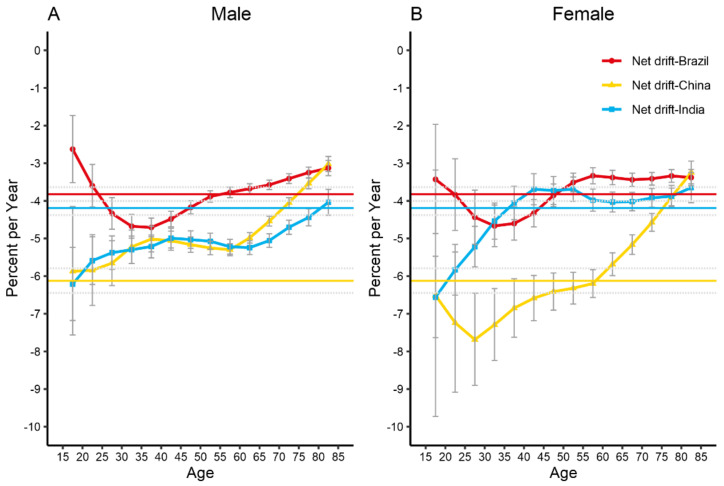
Local drift with net drift values for PUD mortality in China, Brazil, and India by sex during 1990−2019. (**A**) for males; (**B**) for females. Error bars represent the 95% CIs for the local drift values. The straight lines of yellow, red, and blue represent the net drifts of China, Brazil, and India respectively, and gray dashed lines represent their 95% CIs.

**Figure 3 healthcare-11-01085-f003:**
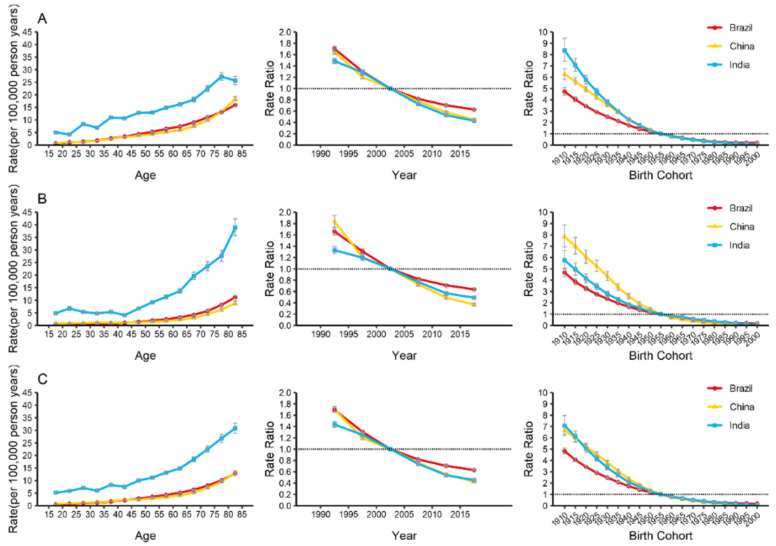
The age, period, and cohort effects on PUD mortality across China, Brazil, and India; (**A**) for males; (**B**) for females; (**C**) for both sexes. Solid lines of red, yellow, and blue represent Brazil, China, and India, respectively. The gray dashed lines represent their reference period (2000–2004) and cohort (1955–1959) respectively. Error bars represent their corresponding 95% CIs.

**Figure 4 healthcare-11-01085-f004:**
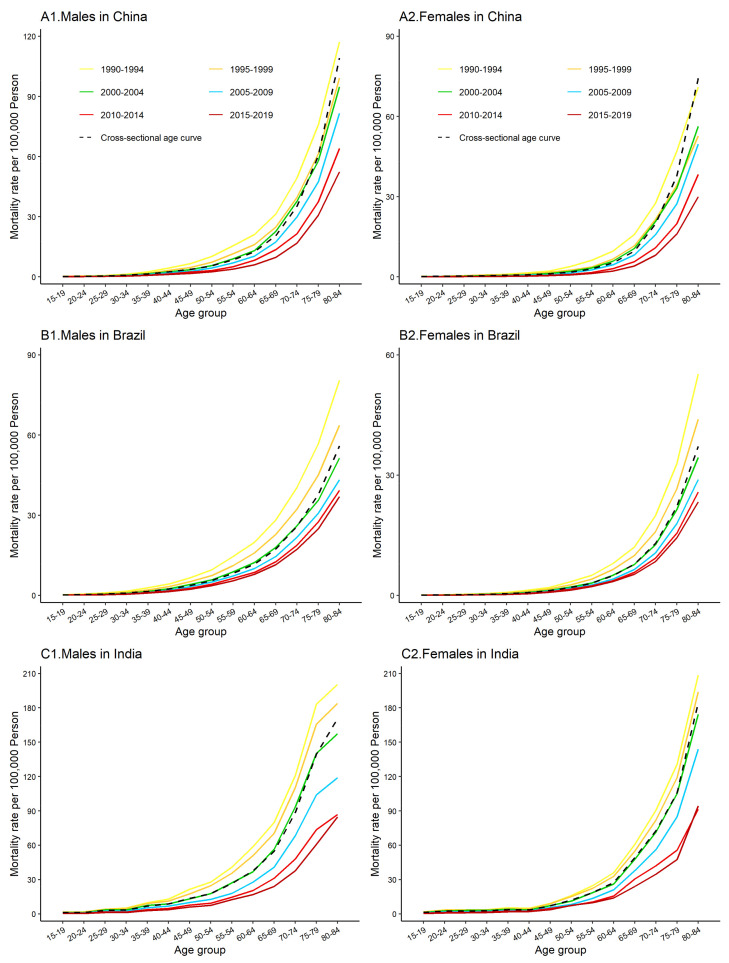
Age–specific mortality rates of PUD by period and sex in China (**A1**,**A2**), Brazil (**B1**,**B2**), and India (**C1**,**C2**) from 1990 to 2019. The research period was divided into 5–year periods, ranging from 1990 to 1994 (median, 1992), 1995 to 1999 (median, 1995), 2000 to 2004 (median, 2002), 2005 to 2009 (median, 2007), 2010 to 2014 (median, 2012), and 2015 to 2019 (median, 2017). After controlling for cohort effects, cross–sectional age curves show the anticipated age–specific rates during the reference period, i.e., 2000 to 2004. (*p* < 0.01 for all).

**Figure 5 healthcare-11-01085-f005:**
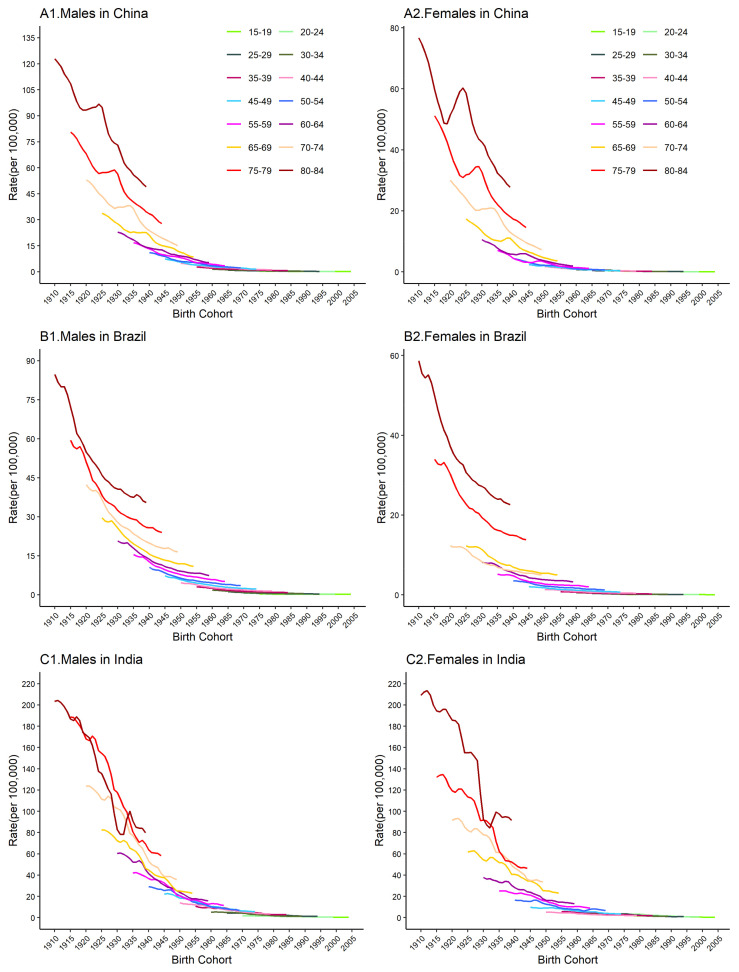
Cohort–specific mortality rate of PUD by age group and sex among China (**A1**,**A2**), Brazil (**B1**,**B2**), and India (**C1**,**C2**) during 1990–2019. Data on PUD mortality were organized into 19 consecutive birth cohorts, ranging from those born in 1910 to 1914 (median, 1912) through 2000 to 2004 (median, 2002), as well as into 5–year intervals starting at age 15 to 19 (median, 17 years) and ending at age 80 to 84 (median, 82 years).

**Table 1 healthcare-11-01085-t001:** Characteristics of PUD deaths in China, Brazil, and India between 1990 and 2019 by sex. ASMR indicates age-standardized mortality rate. (The relative proportion * represents the proportion of all-cause deaths resulting from PUD. The percentage of global represents the percentage of the global population accounted for by China, Brazil, and India.)

		China			Brazil			India	
	Both	Men	Women	Both	Men	Women	Both	Men	Women
ASMR 1/100,000									
1990	7.53	10.02	5.47	6.07	8.23	4.19	20.17	22.78	17.37
2019	2.33	3.37	1.58	2.26	3.04	1.63	6.70	6.87	6.50
Annual percentage change	−3.82	−3.50	−4.05	−3.51	−3.50	−3.41	−4.34	−4.67	−3.92
*p* values	<0.01	<0.01	<0.01	<0.01	<0.01	<0.01	<0.01	<0.01	<0.01
Deaths n × 100									
1990	555	350	205	50	33	17	832	505	327
2019	401	253	148	52	31	21	691	349	342
Relative proportion, % *									
1990	0.66	0.76	0.54	0.51	0.56	0.43	1.01	1.15	0.84
2019	0.38	0.41	0.33	0.37	0.39	0.41	0.74	0.70	0.78
Population n × 100,000									
1990	11837	6102	5735	1488	736	753	8556	4453	4102
2019	14224	7428	6975	2167	1058	1109	13907	7132	6775
Percentage of global									
1990	22.13	22.65	21.59	2.78	2.73	2.83	15.99	16.53	15.45
2019	18.38	18.68	18.09	2.80	2.73	2.87	17.97	18.38	17.57

## Data Availability

All the data we used is publicly available and comes from GBD Results.

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
