# Peer review of "Comparison of Secular Trends in Peptic Ulcer Diseases Mortality in China, Brazil and India during 1990–2019: An Age-Period-Cohort Analysis"

_healthcare, 2023, doi:10.3390/healthcare11081085_

Round 1

Reviewer 1 Report

The article is very interesting on the subject and is well structured and has a well organized progression. The figures are accurate. I would say that a tab is needed on the discussion with the risk factors found in comparison of the countries studied.

Author Response

Thank you very much for your revision, please see the attachment for the detailed reply.

Reviewer 2 Report

1.    Could the author interpret the data? Why and how China, Brazil, and India have an inspiring decline in ASMR of PUD.

2.    The inclusion and exclusion criteria are not clear.

3.    The discussion section is poor. The author has to explain the previous study and discuss that.

4.    I suggest the author make a table and describe the results.

5.       What is the home message of the study.

6.       What is the future study.

Author Response

(The authors gave the same response as above.)

Author Response

(The authors gave the same response as above.)

Reviewer 4 Report

Existing papers have evaluated the incidence of PUD in various countries. It is not clear why this study was conducted in India, Brazil, and China.

A further result of this study is that they focused on Smoking and added more assessments.

However, the relationship between smoking and PUDs, which is the subject of this paper, has been pointed out in the past.

A more detailed evaluation of smoking and eradication seems necessary.

Author Response

(The authors gave the same response as above.)

Round 2

Reviewer 2 Report

Thank you

Author Response

Thank you for your suggestions. We have consulted experts to polish our manuscript and made revisions according to the suggestions of you and other reviewers. Please see the revised manuscript for specific changes.

Reviewer 4 Report

In your paper's concluding paragraph, isn't it an overstatement to say that smoking policies reduced PUD?

Author Response

Thank you for your suggestions.  We have consulted experts to polish our manuscript and made revisions according to the suggestions of you and other reviewers. Please see the attachment for detailed reply content.
